

# Towards accurate quantification of ice content in permafrost of the Central Andes - Part II: an upscaling strategy of geophysical measurements to the catchment scale at two study sites

Tamara Mathys[1], Christin Hilbich[1], Lukas U. Arenson[2], Pablo A. Wainstein[2], and Christian Hauck[1]

[1]University of Fribourg, Department of Geosciences, Switzerland
[2]BGC Engineering Inc., Canada

**Correspondence:** Tamara Mathys (tamara.mathys@unifr.ch)

**Abstract.** With ongoing climate change, there is a pressing need to better understand how much water is stored as ground ice in areas with extensive permafrost occurrence and how the regional water balance may alter in response to the potential generation of melt water from permafrost degradation. However, field-based data on permafrost in remote and mountainous areas such as the South-American Andes is scarce and most current ground ice estimates are based on broadly generalised

assumptions such as volume-area scaling and mean ground ice content estimates of rock glaciers. In addition, ground ice contents in permafrost areas outside of rock glaciers are usually not considered, resulting in a significant uncertainty regarding the volume of ground ice in the Andes, and its hydrological role. In part I of this contribution, Hilbich et al. (submitted) present an extensive geophysical data set based on Electrical Resistivity Tomography (ERT) and Refraction Seismic Tomography (RST) surveys to detect and quantify ground ice of different landforms and surface types in several study regions in the semi-

arid Andes of Chile and Argentina with the aim to contribute to the reduction of this data scarcity. In part II we focus on the development of a methodology for the upscaling of geophysical-based ground ice quantification to an entire catchment to estimate the total ground ice volume (and its estimated water equivalent) in the study areas. In addition to the geophysical data, the upscaling approach is based on a permafrost distribution model and classifications of surface and landform types. Where available, ERT and RST measurements were quantitatively combined to estimate the volumetric ground ice content

using petrophysical relationships within the Four Phase Model (Hauck et al., 2011). In addition to introducing our upscaling methodology, we demonstrate that the estimation of large-scale ground ice volumes can be improved by including (i) non-rock glacier permafrost occurrences, and (ii) field evidence through a large number of geophysical surveys and ground truthing information. The results of our study indicate, that (i) conventional ground ice estimates for rock-glacier dominated catchments without in-situ data may significantly overestimate ground ice contents, and (ii) substantial volumes of ground ice may also be

present in catchments where rock glaciers are lacking.

## 1 Introduction

In many arid and semiarid areas around the world, mountain regions play a significant role for controlling downstream water supply. At high altitudes, runoff is delayed by glaciers, which consequently act as a major source of water for agriculture,



power generation, mining and drinking water during the summer (Urrutia and Vuille, 2009; Salzmann et al., 2013; Rangecroft
et al., 2015; García et al., 2017; Hoelzle et al., 2019). With continued climate change further enhancing the recession of glaciers
globally, their contribution to the summer runoff will eventually decline, whereby the timing of this decline differs globally
based on the region (IPCC, 2019). Therefore, water availability, and specifically its timing, will drastically change. In this
context, the permafrost distribution, the corresponding ground ice content and its degradation is of increasing importance, as
it is currently debated whether permafrost ground ice can be considered as a significant water reservoir and as an alternative
resource of fresh water that could potentially moderate water scarcity during dry seasons in the future (Brenning, 2005; Azócar
and Brenning, 2010; Duguay et al., 2015; Hoelzle et al., 2017; Jones et al., 2019; Liaudat et al., 2020). Thus, there is a pressing
need to better understand i) how much water is stored as ground ice in areas with extensive permafrost occurrence and ii)
how the regional water balance may alter in response to the potential generation of melt water from permafrost degradation.

In the arid and semi-arid Andes, permafrost and specific permafrost landforms such as rock glaciers are known to be well
developed and abundant (Azócar and Brenning, 2010; Jones et al., 2018a, 2019; Schaffer et al., 2019; Masiokas et al., 2020).
Several recent studies attempt to estimate and model water volumes and the potential hydrological significance of rock glaciers
by quantifying their ground ice contents (Azócar and Brenning, 2010; Rangecroft et al., 2015). These studies hypothesise that
a significant amount of water may be stored in the ice-rich layers of rock glaciers, however, those assertions are typically
not supported by in-situ measurements. If present (and future) periglacial protection laws in the South American Andes (e.g.
Chile and Argentina) are to be practically adopted, a more precise understanding of the distribution, ground ice content and
thermal state of permafrost is necessary. Most of the large-scale studies addressing ground ice volumes in rock glaciers are
based on remote sensing data and use empirical rules of thumb to estimate the ground ice content without any ground truthing
as validation. Due to the lack of in-situ investigations and subsurface information, commonly a mean volumetric ice content of
40% to 60% is assumed for the entire rock glacier area (Brenning, 2005; Rangecroft et al., 2015; Bodin et al., 2010; Azócar and
Brenning, 2010; Rangecroft et al., 2015; Jones et al., 2019). This can be problematic as the ground ice content of rock glaciers
has been shown to vary considerably from case to case (Arenson and Jakob, 2010; Hauck et al., 2011; Mollaret et al., 2020;
Halla et al., 2020, Hilbich et al., submitted) and also along longitudinal profiles of a single rock glacier (Jones et al., 2019;
Halla et al., 2020). Furthermore, while rock glaciers are the focus of several studies assessing the hydrological importance of
permafrost in semi-arid regions, knowledge about the permafrost distribution outside of rock glaciers is still extremely limited
(Arenson and Jakob, 2010; Duguay et al., 2015), even though ice-rich ground ice has been shown to be present in areas devoid
of rock glaciers (García et al., 2017; Schaffer et al., 2019). The water equivalent stored in ice-rich permafrost zones can thus
be expected to be significantly higher than calculated from rock glaciers alone (Arenson and Jakob, 2010; García et al., 2017;
Baldis and Liaudat, 2020). Clearly, field based studies are needed to fully assess the partitioning of the different permafrost
landforms to the total ground ice volume stored in permafrost (e.g. Schaffer et al., 2019; Croce and Milana, 2002; Azócar and
Brenning, 2010; Arenson and Jakob, 2010; Monnier and Kinnard, 2013; Duguay et al., 2015; Azócar et al., 2017; Schaffer
et al., 2019; Halla et al., 2020). Moreover, the few existing (statistical) permafrost distribution models of the Central Andes
(and elsewhere) are commonly based on the presence and distribution of active rock glaciers and can therefore not easily be



extended to other types of surface covers (Arenson and Jakob, 2010; Bodin et al., 2010; Azócar et al., 2017; Esper Angillieri,
2017). This bias is caused by the inability of available remote sensing data to detect permafrost from space outside of clear
geomorphic indicators, such as rock glaciers.

In order to remove this bias and estimate the total ground ice volume more accurately, Hilbich et al. (submitted) presented
an extensive geophysical data set from several permafrost regions in the Central Andes of Chile and Argentina consisting
of Electrical Resistivity Tomography (ERT) and Refraction Seismic Tomography (RST) measurements, which can be used
for the detection of permafrost and the quantification of the ground ice content. In the absence of boreholes, geophysical
investigations are a feasible and cost-effective technique to detect and quantify ground ice occurrences within a variety of
landforms and substrates (e.g. Hauck and Kneisel, 2008a; Hilbich et al., 2009, 2011; Monnier and Kinnard, 2013; Mewes
et al., 2017; Pellet and Hauck, 2017; Mollaret et al., 2019; Halla et al., 2020; Mollaret et al., 2020). Several combinations
of geophysical measurements have been used in this context. Monnier and Kinnard (2013) used ground-penetrating radar
to estimate the ground ice content in a rock glacier in the Central Andes of Chile. Halla et al. (2020) used ERT and RST
data in combination with the so-called 4-phase model (Hauck et al. 2011) to calculate the ground ice content within a rock
glacier complex in Argentina. In further studies on rock glaciers in other mountain ranges electromagnetic (Bucki et al., 2004),
gravimetric (Hausmann et al., 2007) and multitudes of geophysical techniques (Maurer and Hauck, 2007; Buchli et al., 2018)
were applied, often in combination with ground truth data from boreholes.

Finally, and in addition to the geophysical surveys and local ground ice calculations themselves, upscaling techniques are
needed to estimate the ground ice content over larger scales than the actual profile lengths of the geophysical surveys, such
as a whole watershed (e.g. Minsley et al., 2012; Hubbard et al., 2013; Dafflon et al., 2017). Upscaling is needed when water
balances and potential future changes on larger spatial scales have to be assessed. In this study we aim to bridge the scale gap
between the individual geophysical profiles and the catchment scale by establishing an upscaling methodology, which is then
applied to two study sites in the Central Andes of Chile and Argentina. The geophysical data and analyses are hereby based
on the companion paper in part I (Hilbich et al., submitted). We present first estimates of total ground ice content (and water
equivalent) of a) a rock glacier dominated and b) a rock glacier free site. We then compare the results of our proposed upscaling
methodology to conventional estimates based on remotely sensed data and the empirical rule of thumb established by Brenning
(2005).

## 2  Study Sites

The field data of the two sites were acquired in summers 2018 and 2019 in the framework of environmental impact assessment
studies (EIA) in mining environments. Profile locations were chosen according to the probable presence of frozen ground, but
also according to easy access and safety within the mines. Locations are therefore not always optimally situated with respect
to representativity in an upscaling context. The local mining context has no further relevance for the scientific content of this



paper and is therefore omitted in the following. The mining infrastructure, however, enabled access to high-altitude permafrost environments and made the collection of validation data possible.

## 2.1 Rock glacier free site - Site D

The rock glacier free site is located between 3800 to 5400 m a.s.l., about 140 km southeast of the city of Copiapò (cf. Site D in Figure 1). The climate of Site D is classified as semi-arid, typical of high altitude Central Andes. Precipitation and 95 relative humidity are low, while solar radiation is high. Mean annual air temperature (MAAT) calculated from a meteorological weather station that was installed on site at an altitude of 5012 m a.s.l. in 2015 is -5.4 °C (for the measurement period of 2015 - 2017). Using long-term representative regional climate stations in the vicinity of the site, a MAAT of -7.3 °C was estimated for the period of 1978 to 2015 (Devine et al., 2019). Mean annual precipitation estimated from the same long-term series is 131 mm, with significant variability (annual values ranging from 0 mm to 738 mm). Site D is characterized by a uniform 100 landscape consisting of a widespread fine-grained sediment cover. Rock glaciers are absent at this study site and geomorphic surface indicators for permafrost mostly consist of widespread gelifluction lobes and in some cases weakly developed patterned ground. The results of the geophysical surveys presented in part I (cf. Hilbich et al., submitted) point to largely homogeneous subsurface conditions with significant ground ice occurrences (mostly in terms of a thin, ice-rich layer varying in thicknesses of approximately 2-5 m). Furthermore, ground ice is expected to be present as well in the highly fractured and hydrothermally 105 altered bedrock at greater depths (Hilbich and Hauck, 2018).

## 2.2 Rock glacier site - Site A

Contrasting Site D, we choose a second field site with an abundance of specific permafrost landforms including rock glaciers and talus slopes. Site A is located in the upper part of the Choapa Valley, about 200 km north of Santiago de Chile (Figure 1) and was previously studied by Monnier and Kinnard (2013). The geophysical measurements are located at an altitudinal 110 range of 3560 to 3850 m a.s.l. The climate of site A is arid with a short rainy season between May and August. The annual precipitation ranges between 200 mm and 800 mm with an average of 334 mm for the period of 1961-90 (Monnier and Kinnard, 2013). MAAT is estimated to be around +0.5°C in 2010 at an altitude of 3700 m a.s.l., which is a clear indicator that permafrost would not form under current climatic conditions and it is naturally in a degrading state. Site A contains various distinctive landforms. The most important periglacial landforms present at Site A are rock glaciers with about 15 rock glaciers that have 115 been identified in the area of interest. In general, variable ground ice contents were found in the different rock glaciers and other landforms of Site A (Monnier and Kinnard, 2013, Hilbich et al., submitted.). Areas outside of rock glaciers are most likely unfrozen and free of permafrost, as seen by the relatively low electrical resistivities found in such areas (Hauck et al. 2017, Hilbich et al., submitted), and confirmed by several test pits.





**Figure 1.** Overview maps of study sites A (rock-glacier dominated site, including various sub-study areas) and D (rock-glacier free site) in the Central Andes of Chile/Argentina. The permafrost zonation index after Gruber (2012) shows the modelled permafrost distribution within the Central Andes. Black lines indicate the position of the geophysical profiles. Map data: Site A: Google, ©2021, Maxar Technologies; Site D: DigitalGlobe, accessed in GlobalMapper V22 (Blue Marble Geographics), date unknown

## 3   Methodology

The data sets used for this study consist of (i) geophysical measurements (Electrical Resistivity Tomography, ERT, and Refraction Seismic Tomography, RST), which are part of the large data set presented in the companion paper of Hilbich et al. (submitted), (ii) ground truth data in the form of test pits, boreholes and natural outcrops (unpublished data provided by BGC Engineering Inc. (BGC)), and (iii) geomorphological maps produced during fieldwork in the framework of this study. All data are then combined within a workflow for upscaling from the individual geophysical profile to the catchment scale (Figure 2).

The upscaling approach is based on the assumption, that the geophysical profiles can be considered representative for larger





areas with comparable near-surface substrate or for similar landforms in similar topo-climatic locations, by this representing comparable permafrost occurrences with similar ground ice contents. The workflow consists of five steps: (1) a background permafrost distribution model, (2) partitioning of the study site into different surface/subsurface classes, (3) quantitative ground ice content estimations for each profile, and (4) a conceptual soil stratigraphy model for each upscaling class. In a final step

(5), the ground ice content estimations are integrated over the whole study area according to the soil stratigraphy model and the respective abundance of each upscaling class. While steps (3) and (4) are closely related to the geophysical data and corresponding ground ice content calculations (see section 3.1) , steps (1) and (2) are more flexible and can be based on different approaches. Any permafrost distribution model may be used to distinguish between permafrost classes and non-permafrost classes (1) and any field information and remote sensing data can be used to classify the surface into upscaling classes (2). In

the following, the different steps are further explained.

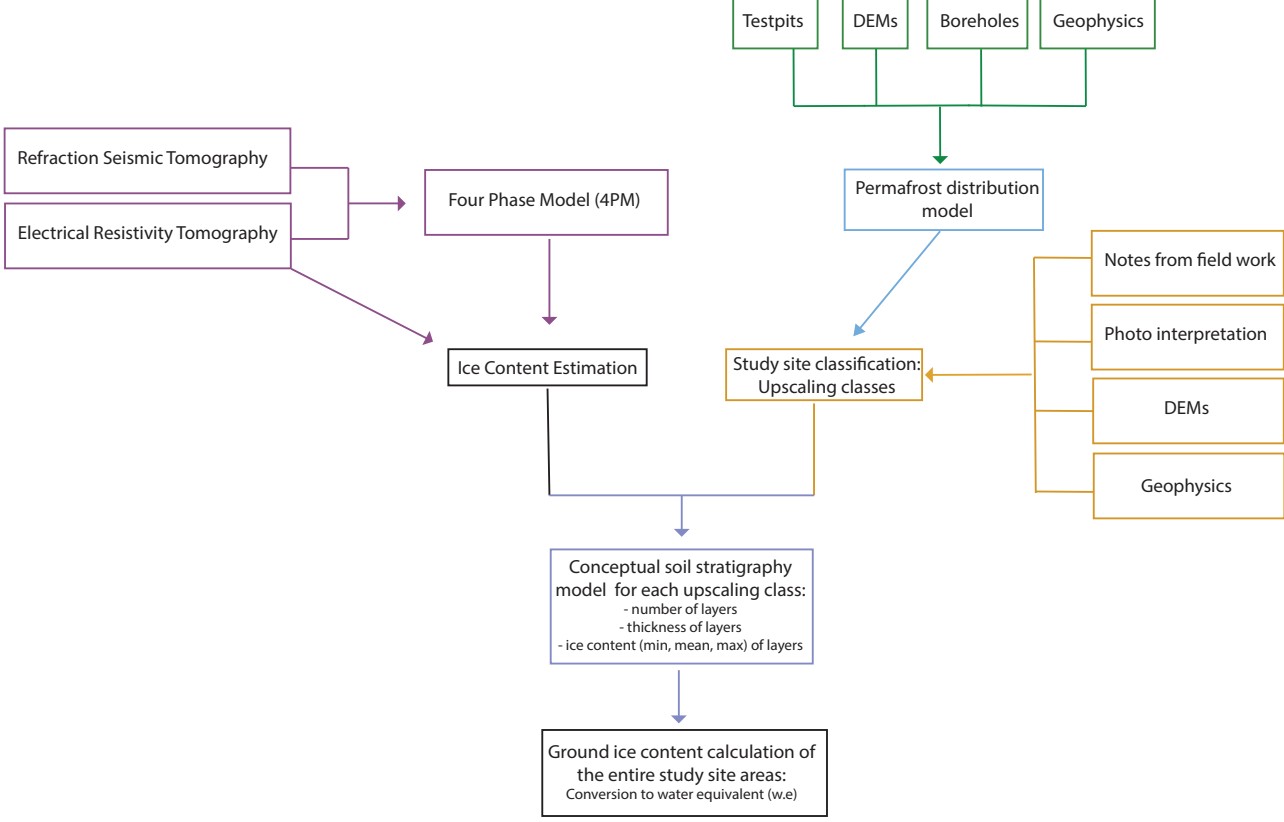

**Figure 2.** Workflow of the upscaling methodology used in this study.



## 3.1 Geophysical datasets and validation

In part I of this study, Hilbich et al. (submitted), provide the details about data acquisition and processing as well as data interpretation in the context of a comparison of several permafrost field sites in the Central Andes. It has to be noted that the geophysical profiles were not planned specifically for the purpose of upscaling to larger regions. However, the extensive

data set allowed us to develop our methodology and apply it to the two field sites described above. Several sources of ground truthing were available for the validation of the interpretation of the geophysical surveys. At Site A, a large set of borehole data exist (Koenig et al., 2019), and the results from the geophysical surveys are in good agreement with the available borehole data (Hilbich et al., submitted; Hauck et al. (2017); Hilbich and Hauck (2018)). At Site D, no geotechnical boreholes exist for calibrating the geophysical lines, however, several test pits were excavated, which largely confirmed the findings from the

geophysical surveys. Additionally, several natural outcrops were present in the vicinity of the profiles and further served as validation data (Hauck et al., 2017; Hilbich and Hauck, 2018). We refer the reader to the companion paper (Hilbich et al., submitted) regarding geophysical data acquisition and processing, as well as the ground ice quantification from co-located ERT and RST tomograms by applying the so-called 4-phase model (4PM, Hauck et al., 2011) .

## 3.2 Permafrost distribution models

For the sake of simplicity and to use our available ground truthing data effectively, we chose a multiple regression approach that was first introduced by Hoelzle (1994) to generate a permafrost distribution model for our field sites. In contrast to e.g. Hoelzle (1994), who used data from the bottom temperature of the snow cover (BTS), we use the results of the geophysical measurements, boreholes and test pits as point input for the response variable Y, indicating likely permafrost occurrence. Elevation and potential incoming solar radiation (PISR) were chosen as the predictor variables (equation 1).

$$y(x) = b_0 + b_1 * elevation(x) + b_2 * PISR(x) \tag{1}$$

The resulting permafrost distribution model delineates areas where permafrost occurrence is probable, possible or unlikely, analogous to similar approaches in earlier studies (Hoelzle, 1994; Gruber and Hoelzle, 2001; Arenson and Jakob, 2010; Esper Angillieri, 2017; Kenner et al., 2019). The main purpose of the model in the context of this study is to provide a coarse overview of the expected permafrost distribution of the study region to be able to directly exclude those areas from further

processing, where permafrost is highly unlikely (cf. yellow areas in Figure 1a, 2a and 3). More complex, statistical permafrost distribution models for the Central Andes have been presented (e.g. Bodin et al., 2010; Azócar et al., 2017), however, they cover only parts of our study sites. A comparison of our model with the model by Azócar et al. (2017) for Site A and with an unpublished model (L. U. Arenson, personal communication), following the approach presented in Arenson and Jakob (2010) for Site D, showed a good agreement regarding the areas where permafrost is estimated to be absent. A comparison between

the permafrost distribution model developed in this study and the global permafrost zonation index (PZI) developed by Gruber (2012) is shown in Figure 3. For Site D, both models predict (high) probability for permafrost occurrence. However, the PZI





distinguishes between areas of very high probability to the West of the site and areas with medium probability for permafrost to the North of the site, whereas our model predicts a high probability for permafrost everywhere. For Site A, the PZI does not cover the entire study site area, probably as a result of the lower altitude. At this site, the PZI predicts a low probability

for permafrost occurrence at all subsites. The model developed in this study also delimits large portions of the study site as unlikely for permafrost. Nevertheless, the model also suggests the possibility of permafrost (light blue areas) for most of the areas, where the geophysical profile lines are located. A higher probability for permafrost is here only modelled for steep and high bedrock slopes.

### 3.3 Study site classification

After excluding areas where permafrost is highly unlikely based on the distribution model, the study sites are subdivided and categorized into different classes, which are hereafter called *upscaling classes*, and which are later used to upscale the estimated ground ice content at geophysical profiles to a larger area. Surface characteristics are feasible data sets for upscaling approaches of geophysical data because the classification can be performed over larger areas by using remote sensing data (e.g. Dafflon et al., 2017). In general, we distinguished different upscaling classes based on factors that are known to strongly influence the

thermal regime of high mountain permafrost. The main classification criterion is the near-surface substrate type (e.g., coarse blocky, bedrock, etc.), as it has been confirmed to be an important factor for the thermal regime and thus permafrost occurrence besides topo-climatic factors. For example, higher ground ice contents have been shown to be present in areas with blocky surface and sub-surface material (e.g. Schneider et al., 2012; Staub, 2015; Gubler et al., 2013). Further, the presence of specific landforms such as active and relict rock glaciers, talus slopes or gelifluction lobes is used as important geomorphic indicator

for permafrost and ice-rich zones, as they have been shown to possess similar ground ice characteristics for a given region (cf. Hilbich et al., submitted). Finally, in order to delimit bedrock areas from areas considered to contain a sediment cover, we use surface slope angles > 30° as a rough indicator for pure bedrock.

In addition, a good general understanding of the study site through field observations and any additional field data will strongly support the classification process, especially in heterogeneous mountain terrain. For Site D, a few general observations

made on site and through the analysis of the geophysical data were used to define the upscaling classes as well as the choice of the conceptual models and the ground ice content estimations described in the following section. For example bedrock, as observed in outcrops during the field campaign, is highly fractured and ice-rich and frozen conditions can be assumed for larger depths and were also observed close to the surface (e.g. shallow active layer) in the various outcrops throughout the study site (Hilbich et al., submitted). Likewise, field observations and the results from the geophysics have shown that the surface as well

as the subsurface conditions are typically homogeneous, and that frozen conditions can be assumed for the entire study site.

The resulting classes are then further divided into smaller areas with presumably similar ground ice contents, based on the results of the geophysical surveys and considering the exposition (aspect) of the area. The result of this classification is shown for Site D in Figure 6. Table 1 lists the factors taken into consideration for the resulting upscaling classes and subdivisions for Site D as well as the available geophysical ground truthing data. For detailed explanations on the ERT and RST tomograms we

refer the reader to Hilbich and Hauck (2018).

**Figure 3.** Comparison of the permafrost distribution model developed in this study (a and c, DEM 5 m resolution) with the PZI model by Gruber (2012) (b and d, based on SRTM30 data set, <1 km resolution). Map data: DigitalGlobe, accessed in GlobalMapper V22 (Blue Marble Geographics), date unknown

## 3.4 Quantification of ground ice contents per class

For each upscaling class, as listed in Table 1, a conceptual soil stratigraphy model was developed to further simplify the complex subsurface conditions into layered models with varying ground ice contents. These conceptual stratigraphies represent a simplified version of representative ERT tomograms, and form the basis for the ground ice content calculation of the entire study site. Where available, the 4PM results are used for the estimation of the ice content of each layer. As the 4PM calculations





| Upscaling class | | landform, substrate/ Geomorphology | Permafrost probability | Aspect | Other Characteristics | Tomogram |
|---|---|---|---|---|---|---|
| 1 | a | Gelifluction lobes | High | Northwest | Fine-grained sediments | D01 |
| | b | Fine sediments, less distinctive gelifluciton lobes | High | Southwest | Fine-grained sediments | D02 |
| 2 | a | Fine Sediments, no special geomorphology | High | Southeast | High altitude | D03 & D04 |
| | b | Fine Sediments, no special geomorphology | High | Southeast | All tomograms in this region contain a conductive layer | D05, D06, D07 & D08 |
| | c | Fine sediments, gelifluction lobes | High | North | Ice-poor bedrock layer beneath sediments | Part of D07 |
| 3 | a | Bedrock (highly-fractured) with shallow sediment cover | High | Southwest | Slope >20° and <30° Ice-rich +/- 2m sediment cover | D09 |
| | b | Bedrock (highly fractured) with shallow sediment cover | High | North | Slope >20° and <30° Ice poor | Based on parts of D07 |
| 4 | a | Bedrock | High | Southwest | Ice-rich | No tomogram |
| | b | Bedrock | High | North | Ice-poor | No tomogram |

**Table 1.** Definition of the upscaling classes used for Site D.

are based on coincident ERT and RST measurements (cf. Hauck et al. 2011) and RST surveys were not conducted at all ERT profile lines in the study area (cf. Hilbich et al., submitted), estimations are based on an interpretation of the resistivity distribution of the subsurface alone, in combination with ground truthing data, wherever RST data are absent. Figure 4 and Figure 5 show examples of such soil stratigraphy models for both study sites and both cases (with and without coincident RST 210 data). The total ground ice content was then calculated using equation (2), which calculates the sum of the ground ice contents $x_i$ within each of the (total number of n) identified subsurface layers (i):

$$\sum_{i=1}^{n} X_i = (thickness * area * ice\,content) \qquad (2)$$

A minimum, mean and maximum ground ice content was considered for each layer in order to account for the uncertainties resulting from the approximation of a homogeneous thickness, area and ice content of each layer. For the (ice-rich) layers we 215 use the same minimum, mean and maximum ice content estimates as applied in the so-called zone-of-interest (ZOI) in part I of this study (Hilbich et al., submitted). Finally, the water equivalent (w.e.) was calculated for both sites assuming an ice density conversion factor of 0.9 g cm$^{-3}$ (e.g. Paterson, 1994). In order to compare the sites and different landforms directly, the resulting ice contents and water equivalents are in a final step expressed as unit per 1 km$^2$.



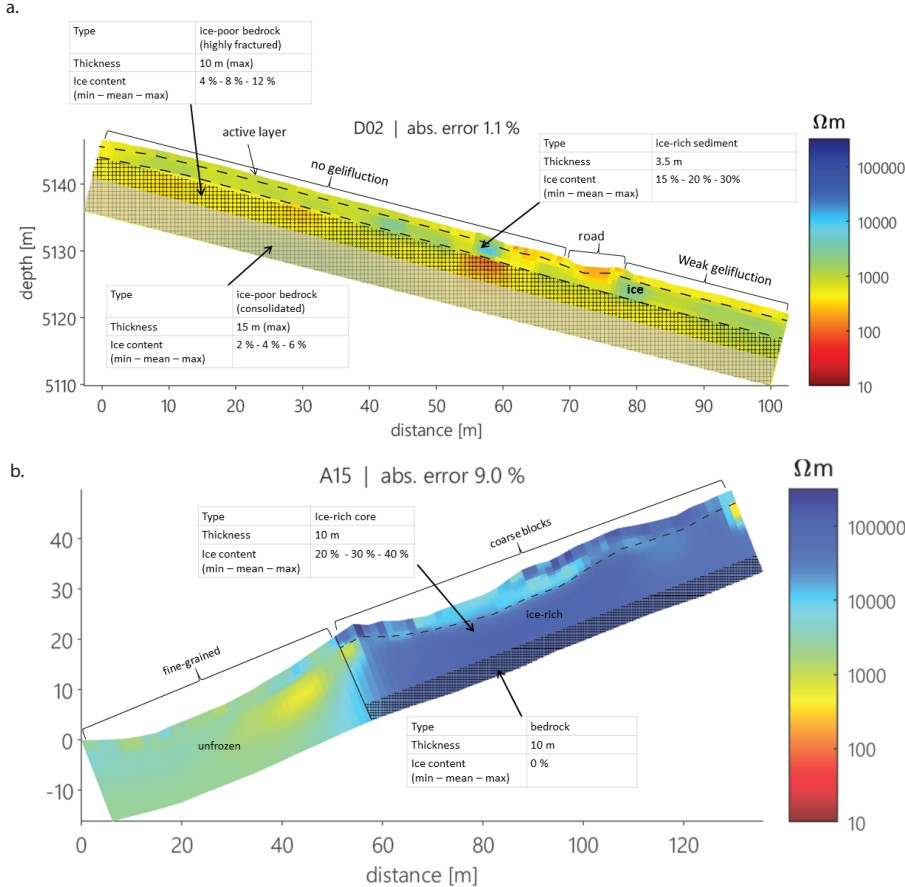

**Figure 4.** Examples of soil stratigraphy models for both study sites (a = Site D, b = Site A), where 4PM results are not available. Here, ground ice contents are estimated for the different layers based on expert knowledge and ground truthing from test pits and outcrops.

.

## 4 Results

The methodology was applied to both study sites, in order to test it for two cases that differ significantly in their surface and subsurface characteristics and corresponding permafrost distribution and conditions. As can be seen from the permafrost distribution models (Figure 3), Site D is located in a zone with high permafrost probability and significant potential for the occurrence of ground ice (Figure 3, 1a,b), despite the absence of ice-rich landforms such as rock glaciers (Hilbich and Hauck, 2018). On the contrary, permafrost ground ice is limited to rock glaciers and possibly talus slopes at Site A (Hilbich et al.,
2018) (cf. Figure 3, 2a,b).

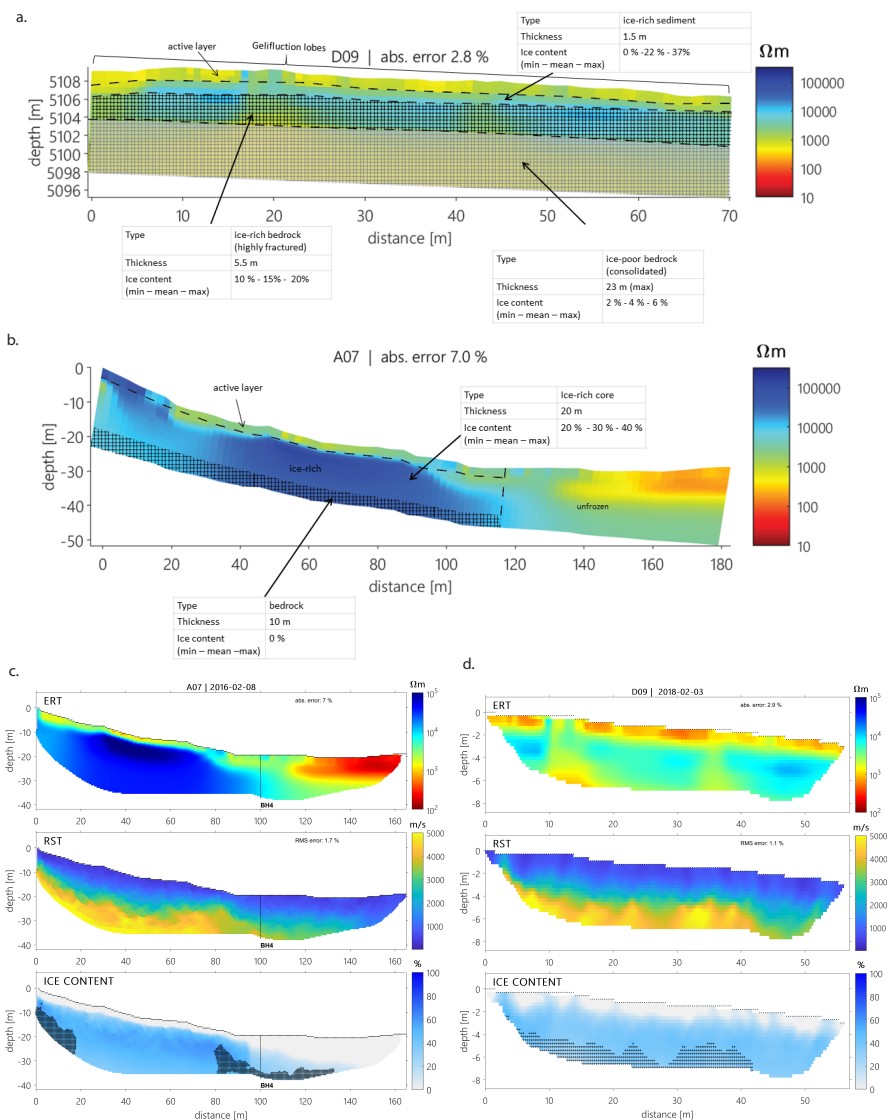

**Figure 5.** Examples of soil stratigraphy models for both study sites, where the 4PM was applied. a and b): soil stratigraphy models, (c and d): corresponding 4PM results (ERT, RST and ice content) (Hilbich et al., 2018). Here, ground ice content estimations stem from the 4PM results: The ice content values for the ice-rich layers correspond to the results from part I of this study (Hilbich et al., submitted). Ice content estimations for layers that are not considered in detail in part I are mean values from the 4PM results (cf. Hilbich and Hauck (2018); Hilbich et al. (2018))

.

For both sites, we calculated total volumetric ground ice estimates (section 4.3) according to two different scenarios with different assumptions regarding the spatial extent of ice-bearing layers for each site. For Site D, the two scenarios differ with respect to their investigation depths:





(S1) Calculation of ground ice content contained in the uppermost 10 meters. This scenario gives a relatively good estimate for the ground ice content in the near-surface layer of the study site, where the reliability of the geophysical surveys is still high.

(S2) Calculation of the ground ice content down to a maximum depth of 30 m. Obviously, in areas with potentially deep permafrost occurrences, the total estimated ground ice content volume depends on the depth considered. This second scenario considers a larger investigation depth, which corresponds to three times the extent of scenario S1 and may be more relevant for modelling purposes in the context of permafrost degradation and its impact on the water balance. Frozen ground conditions to even larger depths can be assumed for Site D, however, uncertainties regarding subsurface conditions (porosity, ice content, etc.) at larger depths are high.

For the rock glacier Site A, where permafrost is limited to rock glaciers and possibly talus slopes, the following scenarios were considered:

(S3) Calculating the ice content of each sub-catchment by only considering the identified rock glaciers.

(S4) As S3, but including talus slopes as additional potentially ice-rich landforms, due to their coarse-blocky surface layer.

In the following sections, we will illustrate the individual steps of the upscaling methodology in detail only for Site D. However, the fundamental differences in ground ice distribution of the sites result in slightly different upscaling approaches for Site A and D. Therefore, we will also shortly explain the most important assumptions made for the ground ice content quantification of Site A.

### 4.1 Permafrost distribution model

The permafrost distribution model developed in the framework of this study mainly served to identify likely unfrozen areas that can be excluded from the overall area for the upscaling (Figure 3). In total, 50 point indicators for permafrost occurrence were used for the regression models. These points correspond to either a location on a geophysical profile or a test pit located at elevations ranging from 4400 m a.s.l. to 5151 m a.s.l and comprise both permafrost and non-permafrost observations. A standard error of 0.58 and an R-squared of 0.6 was calculated. These regression statistics resemble the values used for models in other studies (e.g. Hoelzle (1994)), and are therefore considered to be acceptable. The model indicates a high probability of permafrost for the entire area of Site D, but less probable permafrost conditions at Site A. This result is very similar to the results for the Permafrost Zonation Index (Gruber 2012) (shown in Figure 3 b and d) and an unpublished model of the region (L. U. Arenson, personal communication), which estimates that more than 90 % of Site D is underlain by permafrost. As a consequence of these clear differences between the two sites, the entire area of Site D was considered for the following steps of the methodology, whereas only rock glaciers and talus slopes were considered at Site A.





## 4.2 Classification into upscaling classes

The upscaling classes determined for Site D are listed in Table 1, together with their specific surface characteristics, permafrost
probability, aspect and the tomogram that is representative of each upscaling class. These ERT tomograms can be found in
the Appendix (Figure A1). Further, Site D was split into sub-catchment 1 to the west, and sub-catchment 2 located on the
east (Figure 1a and Figure 3). For the classification, the surface types and landforms were mapped manually based on satellite
images and geomorphological mapping during the field campaign. At Site D they include fine-grained (colluvial) sediments,
gelifluction lobes, bedrock with a sediment veneer (defined as approximately <2 m sediment covering the bedrock), and pure
bedrock. Sub-catchment 1 is characterized by a widespread fine-grained sediment cover, which is further divided into a) the
northwestern slope of the sub-catchment with fine-grained sediment with gelifluction lobes (upscaling class 1a, with D01
serving as reference tomogram), and b) the southwestern slope dominated by fine-grained sediment without gelifluction lobes
(upscaling class 1b, with D02 serving as reference tomogram). Profile D09 serves as a reference in sub-catchment 1 for slopes
steeper than 20° which were classified as bedrock with a thin sediment cover (sediment veneer), defined as upscaling class 3a.
Hereby, a thin sediment layer of 1.5 - 2 m above a highly fractured, likely ice-rich bedrock was observed at an outcrop close
to profile D09. A small area on the south-east of the site is mapped as upscaling class 4a corresponding to pure bedrock (slope
>30°). It is assumed to be ice-rich in the uppermost layer and ice-poor at larger depths, further considering the results of profile
D09.

A large central area of sub-catchment 2 consists of colluvial, fine-grained sediment that does not contain any special land-
form. Profiles D03, D04, D05, D06, D07 and D08 are all located in this area (Figure 6). The results from the geophysical
surveys point to differences between the profiles located at lower elevations (in the center of the sub-catchment) and profiles
located higher up, in steeper slopes. Therefore, this area was further sub-divided into upscaling classes 2a and 2b. Zone 2a
contains the profiles D05, D06, D07 and D08, which are all characterized by a distinctive conductive layer whose thickness
is probably overestimated by the inversion process (cf. Hilbich and Hauck (2018)). Zone 2b hosts the profiles D03 and D04,
which do not contain said conductive layer. At the foot of the local mountain peak (Zones 4a and 4b) gelifluction lobes and sed-
iments (darker color) representing gelifluction morphologies were observed during the field work. Profile D07, which reaches
the lower boundary of these gelifluction lobes, may point to less ice-rich bedrock conditions compared to what has been found
in D09 in sub-catchment 1. Thus, the right-hand part of D07 is taken as a reference for the ground ice calculations for this area,
defined as Zone 2c.

Similar to sub-catchment 1, slopes steeper than 20° were mapped as bedrock with a shallow sediment cover. This concerns
mostly the northern slope to the east. Because of its northern aspect, and resulting higher exposure to solar radiation, this area
is assumed to be less ice-rich than its south/south-western counterpart in sub-catchment 1. The steepest slopes (> 30°) located
at the top of peak on the south-east were mapped as pure bedrock (Zone 4b). Following the argumentation for Zone 3b, this
area is also considered to be less ice-rich than the pure bedrock of sub-catchment 1.

The geomorphological mapping and classification for Site A yielded areas of fine-grained sediments with slopes < 20° (Class
1), bedrock with a shallow sediment cover (Class 2) (slope < 20° and < 30°), and pure bedrock (Class 3) (slopes > 30°), in

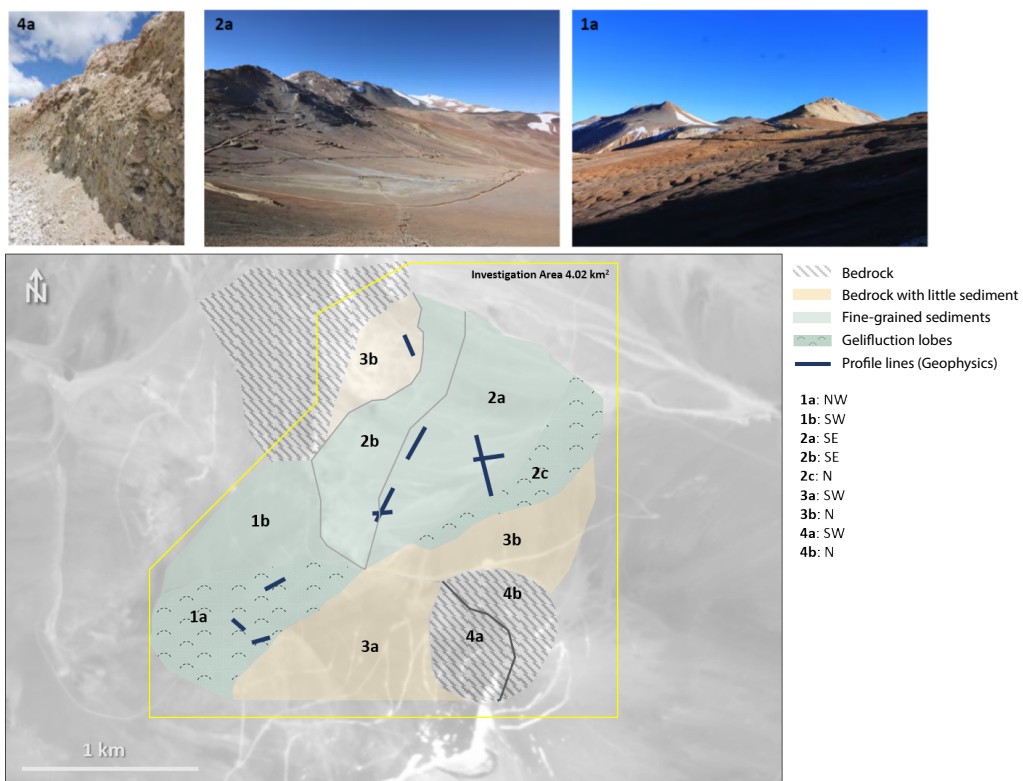

**Figure 6.** Map showing the different upscaling class areas for Site D. The photographs illustrate the general surface types/conditions for some of the most important upscaling classes. Classes 1a, 1b and 3a are situated in sub-catchment 1. Map data: DigitalGlobe, accessed in GlobalMapper V22 (Blue Marble Geographics), date unknown

addition to the mapped rock glacier and talus slope areas for each sub-catchment (cf. Figure 1). However, as the permafrost distribution modelling showed that permafrost can be assumed to be absent outside of rock glaciers and talus slopes at Site A, the resulting upscaling classes containing ice only include rock glaciers (scenario S3), or rock glaciers and talus slopes

(S4) (not shown). Each rock glacier for which the ice content was modelled using the 4PM (cf. Hilbich et al., submitted) is considered an own upscaling class (Class 4x) with a specific subsurface soil stratigraphy and ice content, accounting for their heterogeneity. Rock glacier areas, where no geophysical measurements are available, were first categorized into intact (Class 5, potentially ice-rich) or relict (Class 6, unfrozen), using the rock glacier classification made by Azócar et al. (2017). Talus slope areas were divided into two upscaling classes, differing between north- (Class 7) or south-facing slopes (Class 8), to account

for possibly lower ice contents in north-facing slopes.





| Layer | Max. depth (m) | Ice content (%) |
|---|---|---|
| Active layer | 1 | 0 % (per definition) |
| Sediment | Depending on ERT results | Ice-rich (= max. 85%), or ice-poor (= max. 20%), depending on interpretation of resistivity measurements (ERT) |
| Highly fractured bedrock | 20 | Ice-rich (= max. 12%) or ice-poor (= max. 6 %), depending on interpretation of resistivity measurements (ERT) |
| (more) Consolidated bedrock | 30 | Ice-poor (= max. 6 % ) |

**Table 2.** General assumptions of layers used in the conceptual soil stratigraphy models for the ground ice content calculations for Site D, where (i) no 4PM results exist, (ii) ground truthing data is absent and/or (iii) the investigation depth from the geophysical methods is not sufficient.

### 4.3 Ground ice content quantification

The ground ice content was quantified for each upscaling class based on the conceptual soil stratigraphy models (cf. Figure 4, Figure 5). In cases with insufficient investigation depth of the geophysical surveys, or where certain layer boundaries could not be resolved, additional assumptions had to be made for individual layers of the upscaling classes. Table 2 summarizes the generalized layer structure (active layer, ice-rich sediment layer and the bedrock layer) including maximum thicknesses and mean ice contents for such cases at Site D.

For Site A, ground ice contents for the ice-rich core of rock glaciers (Class 4x) were taken from the 4PM results (cf. Hilbich et al., submitted), where available. The bedrock below this frozen layer is assumed to be unfrozen in the entire study area. Where no geophysical measurements are available for ice content estimations, the ice content of the ice-rich core of intact rock glaciers (Class 5) are derived using average values from Site A for rock glaciers that are situated in similar settings (aspect, altitude, permafrost probability). Relict rock glaciers (Class 6) are considered to not contain ground ice anymore. For scenario S4, talus slopes were furthermore considered to contain possible ice-bearing layers ((Gude et al., 2003; Hauck and Kneisel, 2008b; Mollaret et al., 2020)). Here, we assume lower ice contents for north-facing talus slopes (Class 7) ranging from 0% to 10% than for south-facing slopes (Class 8), where we applied an ice content range of 5% to 20%. These assumptions stem from the hypothesis that talus slopes could potentially store higher ice volumes as a result of their blocky (insulating) surface layer that allows for more effective cooling (e.g. Wicky and Hauck, 2017). However, the geophysical and 4PM results on talus slopes presented in Part I of this study (Hilbich et al., submitted), are not conclusive in terms of ice content. The numbers presented in this part of the study should, therefore, be viewed with care. More research is clearly needed to better estimate the ice content in talus slopes, as the current data coverage is very scarce.

The calculated total ground ice contents per upscaling class were then summed up for the entire considered area and converted into water equivalent per 1 km$^2$ (Figure 7). The total ground ice volumes on the catchment scale calculated for Site D (total area of 4.02 km$^2$) range between 0.0037-0.0095 km$^3$ (= 4.11-10.57 m w.e) for an investigation depth of 10 m (scenario



S1) and 0.0065-0.016 km$^3$ (=7.3-18.4 m w.e) for an investigation depth of 30 m (scenario S2), respectively. These results equal to a water equivalent per km$^2$ ranging from 1.02-2.61 m w.e for a depth of 10 m, and 1.81-4.57 m w.e for the 30 m

investigation depth. For Site A, the total ground ice volumes on the catchment scale (with a total area of 15.32 km$^2$) range between 0.0010-0.0047 km$^3$ (= 1.11-5.2 m w.e) for scenario 3 and 0.0013-0.006 km$^3$ (=1.46 - 6.62 m w.e) for scenario 4. Here, the water equivalent per km$^2$ amounts to 0.10-0.43 m w.e, considering the entire study area (Figure 7). Values for the Site A sub-catchments are in a similar range. Therefore, the permafrost ground ice volumes and corresponding water equivalents are substantially larger at Site D. This suggests, that non-rock glacier catchments may also contain substantial volumes of ground

ice. It is, however, important to note that this ground ice does not necessarily influence the water balance and probably only marginally influences the annual watershed hydrology (see discussion in section 5).

To our knowledge, permafrost ground ice outside of rock glaciers has not yet been quantified or documented in discussions related to permafrost in the Andes before. With the upscaling methodology established in the framework of this study, ground ice content estimations are now possible also outside of rock glacier dominated catchments, and the relative importance of

different permafrost occurrences for total ground ice volumes can be evaluated. For Site D with a total area of 4.02 km$^2$, the ground ice volume estimates are 0.0037-0.0095 km$^3$ for an investigation depth of 10 m (scenario S1) and 0.0065-0.016 km$^3$ for an investigation depth of 30 m (scenario S2), which convert to a water equivalent of 4.1–10.5 m w.e. and 7.3–18.4 m w.e. for the entire study area. The uncertainty range results from the minimum, and maximum ground ice content estimates per soil layer. These results point to substantial ground ice occurrences outside of rock glaciers. In direct comparison to the rock

glacier area of Site A, the total water equivalents calculated for Site D contain only 7-17% and 12-30% (for 10 and 30 m investigation depth) of the water equivalent stored in rock glaciers at Site A (containing a maximum of 9.0 m w.e). However, on the catchment scale, when the total water equivalent of the two catchments is compared relative to the total area per km$^2$, the values for Site D are several times larger (Figure 7). Based on ground ice volume estimates contained in 1 km$^2$ of the defined catchment areas, Site A contains only about 10% of what is stored at Site D for the 30 m depth scenario and about 18% for

the 10 m depth scenario. This is considered reasonable, as most of the catchment area of Site A (outside of rock glaciers) is expected to be unfrozen (permafrost-free).

## 5 Discussion

It is acknowledged, that the absolute numbers of our results should be considered with care, as there are many different uncertainties involved in the various steps of our upscaling methodology:

(i) The estimation of the ice content of the different layers in the conceptual soil stratigraphy models for each site is challenging, as the prescription of a realistic porosity model is a known weakness of the geophysics-based 4-phase model (4PM) (Pellet and Hauck, 2017; Hilbich and Hauck, 2018). This has been addressed with the development of the so-called petrophysical joint inversion (PJI) scheme that is able to invert for porosity in addition to ice and water content (Wagner et al.2019, Mollaret et al. 2020). However, the successful application of the PJI to a large number of geologi-



**Figure 7.** Calculated minimum/maximum water equivalents (m w.e) per 1 km² for the different scenarios (S1 - S4), as well as for rock glaciers and talus slopes at Site A only. Site A was partitioned into smaller sub-catchments as a result of its larger size in comparison to Site D. The partitioning into Sub 1, Sub 2 and Sub3/4 is a function of the spatial "clustering" of geophysical profiles in Site A. Site D was not further subdivided for the calculation of the total ground ice contents/water equivalents as the total area is comparatively small and because the geophysical profiles are located in closer proximity. For S1 and S2, the black bars indicate the uncertainty resulting from using different delineation scenarios for the upscaling classes (see Discussion)

cally and geomorphologically different profiles is still difficult due to convergence problems in the absence of a priori knowledge (see part I, Hilbich et al., submitted). For the calculations of this study, the porosity models were adjusted for each profile based on a) the landform and the surface substrate, b) the interpreted geophysical results, and c) ground truthing information, where available. We assume porosity values around 50% for the uppermost sediment layers at the sediment sites, and around 40% for a surface covered with highly fractured bedrock (with a decreasing gradient for

more consolidated bedrock that is expected underneath). For the rock glaciers of Site A, porosity values > 60 % were prescribed in case of highly resistive layers to allow for supersaturated conditions (cf. Hilbich et al., submitted). As these





values are merely assumptions, porosity estimates could be a large source of uncertainty by leading to either over- or underestimation of the ground ice contents.

(ii) Although being capable of estimating ground ice content from the geophysical results (e.g. Pellet et al., 2016; Halla
et al., 2018; Mollaret et al., 2019; de Pasquale et al., 2020), the success of the 4PM application depends on the available data. The identification of representative ground ice content of the different layers is therefore often not possible from the geophysical results alone. In addition, the 4PM requires co-located ERT and RST measurements, which are not always available in a sufficiently large number to be used in an upscaling approach as presented in our study.

(iii) The conceptual soil stratigraphy models used here assume a constant thickness and ice content for the entire upscaling
class (under the assumption of relatively homogeneous conditions at Site D). However, lateral differences are to be expected, and the derived ground ice content estimates can only serve as rough approximations on the catchment scale.

The main uncertainties of the ice content estimation result from a) the assumptions for min/max ice contents (partly related to uncertainties of the 4PM), and b) from the delineation of the upsaling classes. Especially in catchments, where landforms with clear morphological outlines are missing, the latter may cause substantial uncertainties. To assess the sensitivity of the cal-
culated ground ice contents to the delineation of the upscaling classes, different possible delineation scenarios were compared for Site D, by either using different tomograms as reference for the ice content of an upscaling class, or by assuming more/less sub-classes (i.e. combining similar classes to larger upscaling classes). The resulting ranges of the ground ice contents calculated for each scenario reach from 14 % to 28 % (shown by black bars for S1 and S2 in Figure 7), with lower uncertainty ranges for the 10 m investigation depth. Wherever geophysical data are available in combination with the observations made
during the field campaigns, we can be rather confident about the results for an investigation depth of 10 m (cf. comparison with ground truthing data in part I (Hilbich et al., submitted)), whereas the 30 m depth scenario should be considered with more caution as it is close to the limit of the penetration depth of the available geophysical surveys.

Even if the uncertainties remain considerable, the general permafrost distribution and the intra- and inter-landform het-
erogeneity of the considered catchments is much better captured by geophysical profiles than from remote sensing based
approaches alone. This is especially important for catchments that do not contain any clear geomorphic permafrost indicators, such as rock glaciers, that could be delineated using remote sensing data. The upscaling methodology presented in this paper is a first attempt at estimating ground ice contents on a larger scale than just landforms (e.g. rock glaciers or talus slopes). Based on our findings, further studies should focus on improving the knowledge of the ground ice content distribution in different landforms aside from rock glaciers. Furthermore, the upscaling methodology could be improved by e.g. coupling the geophys-
ical results to a spatially distributed ground thermal model or using machine learning image classification schemes that allow to estimate ground ice contents of larger areas by classifying landforms into potentially ice-rich or ice-poor classes.

Most studies published on estimating the hydrological importance of rock glaciers in the Andes base their estimations of the thickness of the ice-rich layer in rock glaciers on an empirical rule proposed by Brenning (2005) (e.g. Esper Angillieri, 2009; Rangecroft et al., 2015; Jones et al., 2018a, 2019). This empirical relation is based on rock glacier geometry obtained



through geomorphological mapping and relates the thickness of the ice-rich layer to the rock glacier area (Brenning, 2005), without explicitly taking into account the potentially complex spatial variability and dependence on rock glacier kinematics and morphology. As shown by the extensive field-based data presented in the companion paper by Hilbich et al. (submitted), the resulting estimates of the thickness of the ice-rich layer of active rock glaciers is in most cases overestimated when compared to what has been estimated from geophysical surveys and boreholes. For some rock glaciers the empirical rule based estimates

of rock glacier thickness is twice as large as what has been found through the geophysical measurements, where the thickness of a potentially ice-rich core can be delimited with high confidence due to the large resistivity contrasts between the different layers. In consequence, area-thickness based calculations of ground ice volumes are significantly higher than calculations that are based on geophysics. The additional assumption of a general volumetric ice content of 50% certainly adds to the overestimation of the ice volume contained in certain rock glaciers. Hilbich et al. (submitted) found that such previous ground

ice content assumptions for rock glaciers correspond to field-based results only when their ice-rich zone alone is considered.

Figure 8 compares the results of the water equivalents for Site A, scenario S3 (only considering rock glaciers), using our field-based upscaling methodology with the results of the total water equivalents calculated using the empirical approach by Brenning (2005), both, per sub-catchment and for the total study site area. The geophysically based results are again given in a range calculated from minimum, mean and maximum ice content estimations per subsurface layer, whereas for the approach of

Brenning (2005), we used minimum, mean and maximum ice content assumptions of 20%, 50% and 60%, respectively, while using the empirical rule for the estimation of the thickness of the ice-rich layer. As can be seen from Figure 8, the mean water equivalent calculated using the empirical rule is much larger in each case compared to the calculations using the upscaling technique of this study, which is based on the geophysical results. Although acknowledging the fact that simplified empirical rules as the one shown above can be useful tools to cover large and remote areas, they clearly over-generalizes the complex and

heterogeneous subsurface conditions and ground ice contents of rock glaciers, and as they do not recognise natural variability and actual periglacial processes, they should, in general, be improved or complemented with field-based evidence.

Furthermore, our results suggest that also non-rock glacier catchments may contain substantial volumes of ground ice. It is, however, important to note that this water, stored in the ground, does not necessarily influence the annual variability in the water balance, as the presence or absence of a rock glacier in a watershed may only marginally influences the annual

watershed hydrology. Other factors, such as the overall permafrost characteristics, terrain types and precipitation pattern, play more significant roles.

## 6 Conclusion

In this paper we presented an upscaling methodology for geophysical measurements with the aim to characterize the subsurface conditions in high mountain areas with regard to the occurrence of permafrost. Further, the potential ground ice content for

the entire area of two study sites was calculated. The study is based on a series of geophysical measurements (ERT and RST) that were carried out at several study sites in the dry Andes of Chile and Argentina (Central Andes), presented in Hilbich et al. (submitted), of which we apply our methodology to one rock glacier dominated and a rock glacier free site.


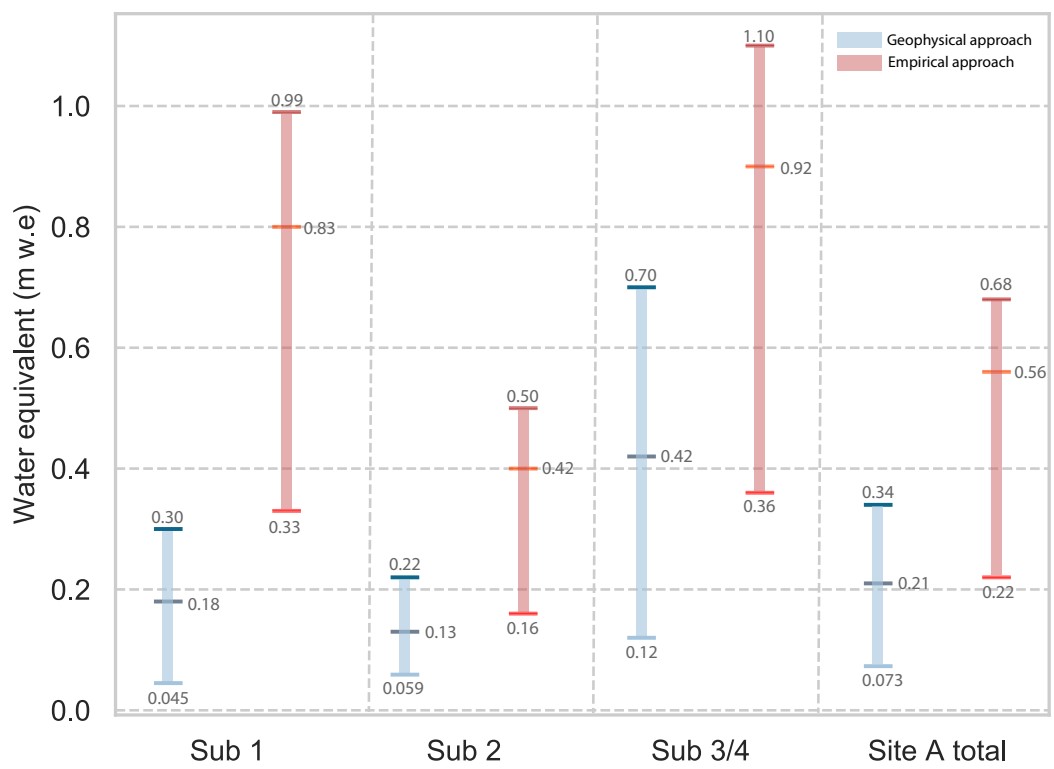

**Figure 8.** Comparison of water equivalent (m w.e per 1 km$^2$) estimations for Site A, scenario S3 (considering only rock glaciers as ice-bearing class), for (i) the geophysically-based upscaling technique (blue bars) presented in this paper and (ii) the empirical rule (red bars) introduced by Brenning (2005).

The upscaling methodology is based on (a) a simple statistical approach that follows previous work on permafrost distribution models, using the geophysical data as permafrost evidence, and (b) a linkage between geophysical results, substrate type and

surface geomorphology to define and delimit areas with presumably similar subsurface conditions (active layer thickness, ground ice content). The so-defined areas were defined as upscaling classes and further used to calculate total ground ice volumes for a larger area. The general approach and strong linkage to the extensive geophysical data sets allowed the estimation of ground ice volumes on the catchment scale. An important advantage of our approach is, that it is not restricted to rock glaciers, as in most studies present to date (Azócar and Brenning, 2010; Rangecroft et al., 2015; Schaffer et al., 2019; Jones

et al., 2018b; Croce and Milana, 2002). We demonstrate, that the estimation of large-scale ground ice volumes can be improved by including (i) non-rock glacier permafrost occurrences, and (ii) field evidence through a large number of geophysical surveys and ground truthing information (where available). Our study presents one of the first estimations of the ground ice volume and

corresponding water equivalents contained in ice-rich permafrost areas in rock-glacier free catchments. It has been shown that such areas can contain a substantial volume of ground ice that should not be neglected in cryospheric, hydrologic, climate or

environmental studies. However, without further investigations and especially the assessment of the temporal scales of runoff contribution of ground ice melt from such extensive permafrost occurrences, its importance for the hydrological cycle cannot be addressed.

**Appendix A: Appendix**

*Author contributions.* TM designed the study, participated in the geophysical campaigns, wrote the major part of the text, and made all fig-

ures. CHi planned, coordinated and participated at the geophysical campaigns, and did the processing of the geophysical data. CHa supervised and contributed to the study design. PW and LA coordinated the environmental impact assessment studies, which included the geophysical campaigns, borehole drilling and the excavation of test pits. They planned and coordinated the field logistics of the geophysical campaigns together with CHi, and provided further background information. All authors contributed actively to the discussion and interpretation of all data sets, and the intermediate and final version of the manuscript.

*Competing interests.* The authors declare that they have no conflict of interest.

*Acknowledgements.* The acquisition of the geophysical data set presented would not have been possible without the valuable support and hard work of numerous field helpers from Chile, Argentina and Switzerland. Therefore we sincerely thank all field helpers for their efforts in the field. The authors also would like to acknowledge the support from various private companies that agreed for having their data published, provided additional information and logistically supported the various field campaigns.



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



**Figure A1.** Inverted tomograms of all ERT profiles for Site A and Site D