# Peer review of "Towards accurate quantification of ice content in permafrost of the Central Andes - Part II: an upscaling strategy of geophysical measurements to the catchment scale at two study sites"

_The Cryosphere, 2021_

## Author Response (AR1)

**RC1:**

General comments:

The manuscript submitted by Mathys et al. describes a new method to extrapolate the quantification of ice contents in permafrost areas based on another paper submitted. By the combination of multiple observations, modelling tools, remote sensing data, they evaluated the ice contents in study sites in the Central Andes. This quantification is helpful for the scientific community to better understand hydrological processes occurring in permafrost-affected catchments.

I think the paper is suitable for publication in The Cryosphere with the condition that the paper explaining the methodology is published as this manuscript fully depends on Hiblich et al. paper. Moreover, I do have some specific comments that may improve the quality of the manuscript.

Specific comments:

Figure 2: It is hard to link the figure with what is described in the text. I would recommend adding some information on the figure to help the reader follow the different steps.

*We adapted the figure and the caption to make this clearer and provide more information to the reader.*

**See adapted Figure 2 with adapted captions on page 7**

L143-145: I suggest to split the sentence into two separate sentences.

*We agree and changed the sentence accordingly.*

**We split the sentence into two separate sentences and split other longer sentences in the manuscript.**

In this way, I found multiple long sentences in the manuscript and splitting them would help the reader.

*Thank you very much for this observation. We revised the manuscript and shorten long sentences in order to improve the readability.*

L223-225: "Figure 3, 1a,b" and "Figure 3, 2a,b" are hard to understand. Please clarify.

*This was an error in the preprint, thanks for spotting it. We changed the text to "Figure 3; a,b" and "Figure 3; c,d". This now also corresponds to the figure labels.*

**The revised sentence is located at L241 and L242.**

Figure 5: I do not see where the text refer to Figure 5. If it is not cited, it should be moved to the Appendix section or removed. Moreover, the figures and the tables within Figure 5 are hard to read.

*Figure 5 is referred to on L209 in the original manuscript. We therefore keep the Figure in the main part of the text but moved the figure closer to the citation and improved the explicit referencing between text and figures.*

**We increase label sizes in Figure 5 to improve readability.**

L253-255: This should go in the Discussion section.

*We would prefer to keep the sentence here, as it is needed to justify our decision of the choice of study areas and their probable permafrost distribution in the following sentences (l255ff: "As a consequence of these clear differences between the two sites, the entire area of Site D was considered for the following steps of the methodology, whereas only rock glaciers and talus slopes were considered at Site A.")*

L312-313: Parentheses are doubled. Please correct.

*Agreed and changed accordingly.*

L332-335: I would suggest to move this sentence to the discussion section.

*We agree and moved the sentence to the Discussion section.*

**We moved the sentence to L418.**

L335-338: These results are already mentioned in the previous paragraph.

*We agree and have deleted the repeating sentences.*

Figure 7: It is not clear that "rock glacier dominated catchment" cover the four groups above.

*We agree and changed the wording to make it clearer to the reader.*

**See adapted Figure 8 on page 19.**

Figure 8: The large uncertainties in the calculations of ice content revealed by this figure are not enough discussed in the discussion section. I would recommend to add information regarding this uncertainty so the reader can understand how the method presented in this paper improves the quantification of ice contents in permafrost areas.

*The uncertainties (bars) shown in Figure 8 origin mainly from the uncertainties regarding the spatial extension of the subsurface ice occurrences and their maximal/minimal values, which are the focus of our geophysical approach (this manuscript, but also the companion paper Hilbich et al., part I). These are explained in lines 406-411 of the original manuscript. By displaying the uncertainties like this, we want to be as clear as possible regarding the chosen geophysical approach. In a purely applied study one would of course narrow down the uncertainty range by choosing a best guess scenario and depth/ice content ranges, which correspond to realistic uncertainties and not to the maximal uncertainties shown in Figure 8. We agree that this has not yet been expressed*

*well enough in the original manuscript and we will therefore discuss the uncertainties in more detail in the discussion section of the revised version.*

**We adapted the discussion in L375 - 419, but especially with regard to the classification process on L 397 - 417:**

IV. *"There are also various uncertainties with regard to the classification of the study area into upscaling classes, especially where clear landforms (e.g. rock glaciers, talus slopes) are absent, as is the case for Site D. The classification process is of rather qualitative nature and strongly relies on field observations and expert knowledge. The large potential uncertainties associated with the classification step are indicated by the uncertainty bars shown in Figure 8. These large error bars origin from the uncertainties regarding the spatial extent of the upscaling classes and corresponding subsurface ice occurrences and their maximal/minimal values. To assess the sensitivity of the calculated ground ice contents to the delineation of the upscaling classes, different possible classification scenarios were compared for Site D. The scenarios were established by either using different tomograms as reference for the ice content of an upscaling class, or by assuming more/less sub-classes (i.e. combining similar classes to larger upscaling classes, or subdividing upscaling classes to smaller ones). The resulting ranges of the ground ice contents calculated for each scenario reach from 14 \% to 28 \% (shown by black bars for SD1 and SD2 in Figure 8, with lower uncertainty ranges for the 10 m investigation depth. The uncertainty range shown in this Figure represents the upper bound. In a purely applied study, one would narrow down this uncertainty range by choosing a best guess scenario for spatial extension, depth of layers and ice content ranges.*

*To resume, the main uncertainties of the ice content estimation result from a) the assumptions for min/max ice contents (partly related to uncertainties of the 4PM), and b) from the classification of the upscaling classes. Especially in catchments, where landforms with clear morphological outlines are missing, the latter may cause substantial uncertainties regarding the spatial extension of the subsurface ice occurrences and their maximal or minimal ice content values. Nevertheless, wherever geophysical data are available in combination with the observations made during the field campaigns, we can be rather confident about the results, e.g. for Site D for an investigation depth of 10 m (cf. comparison with ground truthing data in part I (Hilbich et al., submitted)). The 30 m depth scenario should be considered with more caution as it is close to the limit of the penetration depth of the available geophysical surveys.*

**Also, on L441 – 449:**

*We believe that using field-based evidence (such as geophysical investigations) in combination with large-scale remote sensing data ultimately leads to more realistic ground ice content quantification compared to purely remote sensing approaches. For Site A, we can be confident that ground ice quantification based on the large number of geophysical profiles carried out on the various rock glaciers are more realistic than the pure remote sensing estimates. Furthermore, with the geophysical data presented in Part I of this study (Hilbich et al., submitted) it becomes clear, that rock glacier ice contents may vary significantly from the commonly used 40 - 60 \%. This is why our upscaling approach strongly relies on in-situ data although it means that it is not directly transferable to other catchments where no geophysical data exist. Nevertheless, we believe that field-based data (geophysical or other) are in any*

*case essential for validating remote sensing products or model-based estimates on ground ice contents.*

**RC2:**

This paper aims at developing a methodology to scale local, geophysics-derived estimates of ground ice content to a subcatchment scale. The study is part of a project that uses geophysical data to estimate ground-ice in an area of the Andes. This work is currently under review as the first part of this study.

Similar upscaling attempts have been shown to be successful, but were mostly applied to high-latitude environments, whereas this study is considering a high mountain array. Hence, the authors make use of geomorphological data and field observations.

As the authors demonstrate, estimating ground ice content of high altitude, headwater environments is important to assess groundwater resources further downstream, yet a quantitative estimation of this parameter is difficult. Here, the authors build on geophysical data, presented as Part I of this study, to estimate the ground ice content throughout a wider area. By using various input parameters to classify their sites, the authors are able to provide quantitative estimates of ground ice content. While the approach is very interesting, the classification, which forms an integral part of the study, seems poorly constrained, and mostly qualitative. The authors repeat much detail of the geophysical characterization (which is fair, given that this is the most important data set), there is very little detail on the actual classification. No maps are shown that show the other input parameters, such as slope angle, aspect, geomorphology, or the estimated soil parameters, including locations of soil probing, making it impossible to follow or understand how class parameters vary and how they were decided on. Similarly, it is not clear how the parameters that are critical for the ice content calculations (thickness, area, ice content) were upscaled, or determined, particularly for areas without geophysical data. It would be great to also see those as maps.

*Thank you very much for pointing this out - we completely agree with this suggestion. We will add more maps showing the input parameters, i.e. slope and aspect, to make the classification process clearer. The geomorphology, or rather surface characteristics of Site D is already shown in figure 6 of the preprint. Except for the gelifluction lobes, the geomorphology of this site is very homogeneous, but for more complex study areas, the geomorphological classification would induce larger heterogeneities. We will further add the locations of soil probings and boreholes to the permafrost distribution maps.*

*The parameters that are critical for the ice content calculations (thickness, area and ice content) are determined using the geophysical data. Where no geophysical data is available, we relied on field observations. This will be better discussed in the revised Discussion section (by combining it with changes demanded by Reviewer 1 regarding a better discussion of the uncertainties shown in Figure 8)*

**We adapted Figure 6 (page 16) by adding slope and aspect maps used for the classification. We also added soil probing locations to Figure 3 (page 9). Furthermore, we added another new Figure (Figure 7, page 18), which shows the assumed ice content soil stratigraphies for each upscaling class together with the resulting min, mean and max water contents calculated for each class. This new figure also shows the distribution of calculated water equivalents in the study site.**

**We adapted the discussion in L375 - 419, but especially with regard to the classification process on L 397 - 417:**

*V.  "There are also various uncertainties with regard to the classification of the study area into upscaling classes, especially where clear landforms (e.g. rock glaciers, talus slopes) are absent, as is the case for Site D. The classification process is of rather qualitative nature and strongly relies on field observations and expert knowledge. The large potential uncertainties associated with the classification step are indicated by the uncertainty bars shown in Figure 8. These large error bars origin from the uncertainties regarding the spatial extent of the upscaling classes and corresponding subsurface ice occurrences and their maximal/minimal values. To assess the sensitivity of the calculated ground ice contents to the delineation of the upscaling classes, different possible classification scenarios were compared for Site D. The scenarios were established by either using different tomograms as reference for the ice content of an upscaling class, or by assuming more/less sub-classes (i.e. combining similar classes to larger upscaling classes, or subdividing upscaling classes to smaller ones). The resulting ranges of the ground ice contents calculated for each scenario reach from 14 \% to 28 \% (shown by black bars for SD1 and SD2 in Figure 8, with lower uncertainty ranges for the 10 m investigation depth. The uncertainty range shown in this Figure represents the upper bound. In a purely applied study, one would narrow down this uncertainty range by choosing a best guess scenario for spatial extension, depth of layers and ice content ranges.*

*To resume, the main uncertainties of the ice content estimation result from a) the assumptions for min/max ice contents (partly related to uncertainties of the 4PM), and b) from the classification of the upscaling classes. Especially in catchments, where landforms with clear morphological outlines are missing, the latter may cause substantial uncertainties regarding the spatial extension of the subsurface ice occurrences and their maximal or minimal ice content values. Nevertheless, wherever geophysical data are available in combination with the observations made during the field campaigns, we can be rather confident about the results, e.g. for Site D for an investigation depth of 10 m (cf. comparison with ground truthing data in part I (Hilbich et al., submitted)). The 30 m depth scenario should be considered with more caution as it is close to the limit of the penetration depth of the available geophysical surveys.*

**Also, on L441 – 449:**

*We believe that using field-based evidence (such as geophysical investigations) in combination with large-scale remote sensing data ultimately leads to more realistic ground ice content quantification compared to purely remote sensing approaches. For Site A, we can be confident that ground ice quantification based on the large number of geophysical profiles carried out on the various rock glaciers are more realistic than the pure remote sensing estimates. Furthermore, with the geophysical data presented in Part I of this study (Hilbich et al., submitted) it becomes clear, that rock glacier ice contents may vary significantly from the commonly used 40 - 60 \%. This is why our upscaling approach strongly relies on in-situ data although it means that it is not directly transferable to other catchments where no geophysical data exist. Nevertheless, we believe that field-based data (geophysical or other) are in any case essential for validating remote sensing products or model-based estimates on ground ice contents.*

These limitations of the current manuscript makes it difficult to understand what the benefit of the approach is. Comparing Figures 7 and 8, the shown difference between the geophysical based estimate and the empirical approach, could well fall within the uncertainties introduced by using different classifications. Given the strong reliance on

field observations, it is also questionable whether similar approaches could be used more widely to estimate ground ice contents.

*A similar comment was made by Reviewer 1 regarding the shown uncertainties in Figure 8. We agree that shown like that, the benefit of our approach does not become evident and we will try to improve our argumentation (i.e. that our field-based approach does significantly improve the ice content quantification) in the revised version. It is also worth noting that field observations are an important aspect when characterising the ground ice content of a watershed (see discussion of Part I of this two-part contribution), i.e. we do not promote that the upscaling should be done using geophysical investigations only.*

*As a direct reply to the reviewer comment above, we would first like to state that Figure 7 and Figure 8 do not show the same information. In Figure 7, we compare the water equivalents calculated using the upscaling methodology established in the paper per km² for both study sites. It shows that the calculated ground ice content of the site without rock glaciers (Site D) is larger than the ground ice content per km² of the rock glacier catchment. The uncertainties are only shown for Site D, because the classification for this catchment is much more ambiguous due to the lack of clearly definable landforms (such as rock glaciers, talus slopes, etc.). Figure 8, on the other hand, shows a comparison for the water equivalent values calculated for Site A with the approach from this study and the empirical approach presented by Brenning (2005). This is mainly to illustrate that the empirical approach, which is not based on any field data (geophysical data or any other investigations), most likely overestimates the water equivalent values stored in rock glaciers. Because of the vast geophysical data that we have from this study site on many of the rock glaciers, we are confident that our field-based estimates are more realistic. But we agree that the large (and rather theoretical) uncertainty bars do not express this well enough. We will revise the uncertainty discussion in the Discussion Section accordingly (see also our response to a similar comment by Reviewer 1).*

*Furthermore, we would like to highlight that a clear benefit of our approach is that it is not limited to rock glaciers alone but also includes other landforms or permafrost areas in general that may contain ice-rich permafrost, such as Site D. We show that the water equivalent stored in such areas may potentially be significant for further hydrological research. We will also highlight this better in our revised version.*

*Finally, we agree that we strongly rely on geophysical field observations, supported by in-situ ground truthing data, which may make the approach not directly transferable to any other study site, where no such data exist. However, we believe that field observations are essential to understand the ground ice content distribution of catchments exactly because they cannot be inferred from remote sensing data alone, not even for rock glaciers where at least the area can be delimited more or less clearly from satellite pictures. With this study, we want to emphasise this point and warn the scientific community to rely solely on large-scale remote sensing data when quantifying ground ice content in permafrost regions.*

**This is also addressed in the revised Discussion, more specifically on L441-449.**

Next to those rather major comments, please find below some more minor comments:

Line 14-15: I don't think that an abstract should contain references, and I wonder whether the detail on the geophysics is actually needed here, as this paper focuses on the upscaling, not the geophysics.

*We agree and will modify the abstract accordingly.*

**The abstract of the revised manuscript is shortened. We deleted the sentence:**

*"Where available, ERT and RST measurements were quantitatively combined to estimate the volumetric ground ice content using petrophysical relationships within the Four Phase Model (Hauck et al., 2011)."*

Line 89: In a previous sentence you mention that line locations were planned based on "safety within the mines". Does this mean that the chosen sites are active mining sites, and hence not in their natural state? That would make upscaling to natural systems impossible. According to Fig. 1, sub area 1 seems to be located within active mining, whereas others seem outside. I think some more detail is needed here on what the impact of mining on the chosen sites is to justify that mining has no impact on the results.

*Site D is not an active mining site. Here, the impact of the mine is limited to the construction of roads (which facilitated access for us). Site A is an active mining site. However, for this study we only used the geophysical profile results from areas that are undisturbed and away from any disturbances, meaning that some of the profiles located at site A, Sub1, were not considered in this manuscript as they are located on mining waste rock material. We only considered profiles that are not impacted by the mining activities. We will add clarification on the impact of the mining activities on the upscaling process in the revised manuscript.*

**We added the following sentence (L88-91):**

*" Site D is still in an explorative phase, and thus not an active mining site, where impacts are limited to the construction of roads. Site A surrounds an active mining pit. However, we only consider geophysical profile results from areas that are far away from any disturbances, or are only affected by small scale surface disturbances, such as access roads or drilling platforms."*

Line 126: "comparable near-surface substrate [...]" This is a critical assumption for the upscaling, yet the authors do not provide information on the geology and the variability of subsurface properties.

*We agree that some general information on the geological setting would be helpful. We will include short geological descriptions of both study sites in the site description section. However, we did not use a detailed geological map as an input parameter and we propose to omit a geological map. However, we will add a reference to available geological maps so that interested readers may have a look at the geology.*

**We added general information to the geology of both study sites. For Site A, see L126-129:**

*The geology of Site A is characterized by volcanic (both intrusive and extrusive) as well as sedimentary rocks aging from the Middle Triassic until recent. A large part of the study area has been mapped as quartz-diorites and andesites. Quaternary deposits consist of fluvioglacial sediments and morainic deposits. Similar to Site D, several faults associated with alteration and mineralization are located at Site A (Tapia et al., 2016).*

**For Site D, see L105-112:**

*The main host rock at Site D consists of Late Paleozonic (Permian - Triassic) rhyolites and andesites, which are overlain by Jurassic and Cretaceous sediments and conglomeratic clastic rocks that are strongly silicified and altered. Furthermore, Site D is located on a large hydrothermal alteration zone characterized by high sulphidation epithermal alterations and porphyry alterations. The bedrock is in general highly fractured as a result of the numerous fault systems that cross the area (Devine et al., 2019). The results of the geophysical surveys presented in part I (cf. Hilbich et al., submitted) point to largely homogeneous subsurface conditions with significant ground ice occurrences (mostly in terms of a thin, ice-rich layer varying in thicknesses of approximately 2-5 m). Furthermore, ground ice is expected to be present as well in the highly fractured and hydrothermally altered bedrock at greater depths (Hilbich and Hauck, 2018).*

Line 154: Potential incoming solar radiation: How and based on what did you calculate that?

*Potential incoming solar radiation was calculated using ArcGis Pro's "Area Solar Radiation (Spatial Analyst) tool, which uses a DEM as input. The tool also includes the latitude of the sites for calculations of solar declination and solar position in order to derive PISR. A corresponding sentence will be included in the revised manuscript.*

**We added this information at L164-166.**

Line 155: Equation for estimating permafrost occurrence: It would be good to show a figure that shows the data and model fit, and also details the parameters of the model.

*We will look into this comment and decide whether to add a figure as a supplement.*

**We decided to omit plotting the data and model fit. The permafrost distribution models developed for this study are only used as a rough first overview of the study site and, therefore, we believe it is sufficient to express the model fit by giving $R^2$ values and standard deviation (L169 – 170) for both study sites. The values found in our study closely resemble values found in earlier studies using similar approaches.**

Line 172-173: what do you mean by "high bedrock slopes"?

*What we meant here was bedrock at the highest altitudes (> 3800 m a.s.l) of the study site. We modified this sentence accordingly.*

**This is clarified on L191-192.**

Line 186-187: On what data is this threshold based on?

*This is mostly based on field observations and mapping, where in areas >30° (consulting a slope map based on a DEM), no sediment was observed. A corresponding sentence was added in the revised manuscript.*

**The following sentence was added at L276-277:**

*Profile D09 serves as reference for slopes steeper than 20° which were classified as bedrock with a thin sediment cover (sediment veneer) based on field observations and defined as upscaling class 3a.*

Line 197-199: Although you describe the input parameters, there is no clear methodology described here on how you define the classes. This needs more detail and justification.

*We agree that the justification of our methodology could be explained more clearly. Reviewer 1 also commented that additional explications to Figure 2 (upscaling approach) should be given. We addressed these concerns by improving Figure 2, and by adding new maps that show the spatial distribution of input parameters for the class definition (see also our response to one of the major comments) We will also further clarify in the revised manuscript how the classes are defined.*

**This is discussed already for the first comment (see above). We adapted Figure 2, added slope and aspect maps used for the upscaling class delineations to Figure 6.**

Line 202: Given that soil properties will also impact on the ground temperature distributions, shouldn't the soil stratigraphy be an input to the classification?

*The soil stratigraphy, as known from the geophysical surveys and the test pits, is used as qualitative input to the classification, because if two geophysical surveys showed the same stratigraphy this information is used to classify them into the same upscaling class. However, as soil stratigraphic information is usually very sparse (in our case the geophysical profiles, boreholes and test pits) it cannot be used as explicit input parameter continuously over larger areas.*

Line 295 - 298: Given that the scope of your work is upscaling, why do you distinguish areas where geophysical data has been acquired and areas where this has not been done?

*For rock glaciers located in the catchment for which we don't have any geophysical data, we upscale data from rock glaciers with similar altitude and aspect, where ERT and RST measurements are available for the ice content quantification. For rock glaciers where geophysical data exist, we can use the calculated ice contents directly. We will explain this better in the revised version.*

**Since the upscaling in the scope of this work is done on a rather small study site scale, we distinguish between areas where geophysical data is available and areas where geophysical data is not available because we take most of the information for the soil stratigraphies and ground ice contents from the geophysical profiles. Where no profiles are present, more assumptions have to be taken.**

Line 320 - 321: It would be great to see the estimated ground ice content as a map.

*We agree and we will provide a figure showing the estimated ground ice content for each upscaling class of site D together with the soil stratigraphy conceptualizations for each class.*

**We created a new Figure 7 on page 20, which shows ice content soil stratigraphies (min, mean and max estimates) together with the water equivalents calculated for**

**each upscaling class and it's distribution with a map showing the m w.e. distribution for Site D.**

Discussion: The discussion on the geophysical results should be mentioned, but not in that much detail, as it should be part of Part 1 of this study. The uncertainty in the classification is of greater importance.

*We will discuss the classification uncertainty in more detail. This was also a comment from reviewer 1. We will also go through the text again and remove parts where the geophysical results are described in too much detail.*

**The classification uncertainty is now discussed more clearly in L397 - 409.**

Line 390: I don't think that this is necessarily an image classification problem. But you could use machine learning to exploit relationships between surface and subsurface parameters.

*Thank you very much for this comment; that was also what we meant – we will rephrase the sentence accordingly*

**The rephrased sentence can be found at 471-473.**

Figure 4: You prescribe ice-poor bedrock of D02 with an ice content of 4%, and bedrock of A15, which is overlain by ice-rich material with an ice content of 0%. How did you define that? Similar for Fig. 5, where the 4PM model indicates higher ice contents. Does the bedrock geology play a role in your definition of the ice content? If so, shouldn't this be an input to the classification?

*The value for ice content in bedrock is based on field observations and the geophysical results. For site D, the bedrock is highly hydrothermally altered and fractured, which allows for larger pore spaces. We observed at several bedrock outcrops that the bedrock pore space was actually filled with ice. Therefore, we assume an ice content of 4 % for the bedrock at this site. Site A is located at much lower elevation and the geophysical results as well as the borehole data indicate that there is no ground ice present outside from rock glaciers. We conducted a geophysical profile located on bedrock (with a thin sediment veneer), which suggests no ground ice. Therefore, we assume an ice content for bedrock of 0%. So, the bedrock geology does play a role and we will explain this more clearly (and more generally) in the revised version.*

**We addressed this comment by including short geological descriptions for each study site. For Site A, see L124-127 and for Site D L104-111.**